# Distinctive phenotypes and functions of innate lymphoid cells in human decidua during early pregnancy

Oisín Huhn[1,2,3,4,13], Martin A. Ivarsson[4,5,13], Lucy Gardner[4], Mike Hollinshead[4], Jane C Stinchcombe[6], Puran Chen[5], Norman Shreeve[1,2], Olympe Chazara[2,4,7], Lydia E. Farrell [2,4], Jakob Theorell[8,9], Hormas Ghadially[3], Peter Parham[10,11], Gillian Griffiths [6], Amir Horowitz[12], Ashley Moffett [2,4], Andrew M. Sharkey [2,4]* & Francesco Colucci [1,2]*

During early pregnancy, decidual innate lymphoid cells (dILCs) interact with surrounding maternal cells and invading fetal extravillous trophoblasts (EVT). Here, using mass cytometry, we characterise five main dILC subsets: decidual NK cells (dNK)1–3, ILC3s and proliferating NK cells. Following stimulation, dNK2 and dNK3 produce more chemokines than dNK1 including XCL1 which can act on both maternal dendritic cells and fetal EVT. In contrast, dNK1 express receptors including Killer-cell Immunoglobulin-like Receptors (KIR), indicating they respond to HLA class I ligands on EVT. Decidual NK have distinctive organisation and content of granules compared with peripheral blood NK cells. Acquisition of KIR correlates with higher granzyme B levels and increased chemokine production in response to KIR activation, suggesting a link between increased granule content and dNK1 responsiveness. Our analysis shows that dILCs are unique and provide specialised functions dedicated to achieving placental development and successful reproduction.

[1] Department of Obstetrics and Gynaecology, University of Cambridge School of Clinical Medicine, National Institute for Health Research Cambridge Biomedical Research Centre, Cambridge CB2 0SW, UK. [2] Centre for Trophoblast Research, University of Cambridge, Cambridge CB2 3EG, UK. [3] AstraZeneca, Oncology R&D, Granta Park, Cambridge CB21 6GH, UK. [4] Department of Pathology, University of Cambridge, Cambridge CB2 1QP, UK. [5] Center for Infectious Medicine, Department of Medicine Huddinge, Karolinska Institutet, Karolinska University Hospital, Stockholm, Sweden. [6] Cambridge Institute of Medical Research, University of Cambridge, Cambridge, UK. [7] Centre for Genomics Research, Discovery Sciences, Biopharmaceuticals R&D, AstraZeneca, Cambridge CB4 0WG, UK. [8] Autoimmune Neurology Group, Nuffield Department of Clinical Neurosciences, University of Oxford, Oxford, UK. [9] Department of Clinical Neurosciences, Karolinska Institutet, Stockholm, Sweden. [10] Department of Structural Biology, Stanford University School of Medicine, Stanford, CA 94305, USA. [11] Department of Microbiology and Immunology, Stanford University School of Medicine, Stanford, CA 94305, USA. [12] Precision Immunology Institute, Tisch Cancer Institute Icahn School of Medicine at Mount Sinai, 1425 Madison Avenue, New York, NY 10029, USA. [13]These authors contributed equally: Oisín Huhn, Martin A. Ivarsson. *email: as168@cam.ac.uk; fc287@medschl.cam.ac.uk

n humans, the blastocyst implants into the mucosal lining of the uterus, the endometrium, that is transformed into decidua under the influence of progesterone. Interactions between fetally-derived placental trophoblast cells and decidua are critical to placental development and poor placentation is associated with pregnancy complications such as pre-eclampsia and fetal growth restriction[1]. Decidualisation involves all the elements of the mucosa—glands, arteries, immune and stromal cells[2]. The most abundant leucocytes populating first trimester decidua are ILCs with decidual NK cells (dNK) accounting for up to 70% of decidual leucocytes[3]. In addition to dNK, the other major ILC subsets, ILC1s—including intra-epithelial ILC1s, (ieILC1), ILC2s, ILC3s, and LTi-like cells, have all been described within the decidua but there is still no consensus about their exact phenotypic profiles and functional roles[4–8].

dNK have been typically defined as lineage negative (Lin-) CD56superbright cells, distinct from peripheral blood NK cells (pbNK) and other tissue-resident NK cells (trNK)[9,10]. Cytolytic activity towards standard NK targets, such as K562 is weak and IFNγ secretion only found after stimulation with IL-15 or IL-2[11,12]. Crucially, dNK have never been shown to kill normal trophoblast and available evidence points to them playing a physiological role, acting on extravillous trophoblast (EVT) and maternal decidual cells through the production of factors such as GM-CSF and XCL1[13–17]. More recently, the considerable heterogeneity in decidual ILCs is revealed from single cell RNA sequencing (scRNAseq) analysis of isolates of first trimester decidual cells[8]. The RNA profiling suggests there are three main dNK subsets (dNK1-3) together with proliferating NK cells (dNKp) and an ILC3 population (dILC3). Compared with dNK2 and dNK3 cells, dNK1 cells have large granules with increased expression of perforin, granzymes as well as Killer-cell immunoglobulin-like receptors (KIR), CD94/NKG2A and LILRB1. Immunogenetic and functional studies suggest binding of these NK receptors (NKR) to HLA class I ligands, HLA-C, HLA-E, and HLA-G on EVT, triggers responses in dNK that regulate maternal spiral artery remodelling to ensure adequate blood supply to the growing fetus[18–23].

Other evidence suggests that NK cell granules might play a role in cytokine production. In pbNK from healthy individuals, cells with larger granules are not only better killers, but also produce more cytokines[24]. pbNK from patients with Chediak-Higashi syndrome (CHS) possess NK cells with fewer, larger granules that do not polarise towards canonical NK cell target cells compared to normal pbNK[25,26]. Despite this, their cytokine production is more efficient[27]. These results have resonance with the paradoxical findings for dNK that are poorly cytotoxic yet possess large granules filled with perforin and granzymes.

ILCs exhibit considerable heterogeneity and flexibility to differentiate from one type to another, necessitating detailed characterisation within each tissue type[28]. To describe in detail the phenotype and function of dILCs, we have developed a mass cytometry panel and stained matched blood and decidual leucocytes from first trimester pregnancies. Here, we define protein markers that distinguish dILC subsets with distinct roles in the decidua. There are three major subsets of dNK as well as other uterine ILCs. We show that dILC populations are highly diverse and display decidual-specific phenotype and functions. dNK granule content and organisation is distinct from that of pbNK. We also demonstrate that KIR acquisition by dNK is associated with increased granule content at steady state and increased functionality upon cross-linking of an activating KIR. This comprehensive analysis of the phenotype and functional characteristics of dILCs reveals their unique features in this decidual environment dedicated to successful reproduction.

## Results

**Decidual ILC subsets identified by mass cytometry.** Initially, we set out to characterise the heterogeneity of matched cryopreserved first trimester decidual mononuclear cells (dMCs) and PBMCs using a 41-marker mass cytometry panel (Fig. 1a and Supplementary Table 1). As this is the first study of dMCs using mass cytometry, we compared a subset of our panel of antibodies with results generated by conventional flow cytometry. All antibodies behaved comparably across both platforms (Supplementary Fig. 1A, B, Supplementary Table 2). Few previous studies have used cryopreserved dMCs and thus we also compared the staining profiles for fresh and cryopreserved dMC samples by mass cytometry (Supplementary Fig. 2A, B, Supplementary Table 3). Although similar staining profiles were obtained, certain markers were more sensitive to freezing and thawing such as CD49a and there was also variation between donors (Supplementary Fig. 2B, C).

To analyse all the ILC populations in blood and decidua, Lin-mononuclear cells were identified as CD45+CD3-CD19-CD14-HLA-DR- (Supplementary Fig. 3). Their phenotypic differences were visualised by generating a t-distributed stochastic neighbour embedding (tSNE) landscape using all markers not previously included in upstream gating (Fig. 1b and Supplementary Table 1)[29]. We could determine whether certain regions of this phenotypic space are enriched with cells from the blood or the decidua by using a nearest neighbour-based method. In the tissue probability plot, cells surrounded by decidual cells are coloured blue and those surrounded by peripheral blood are coloured red (Fig. 1c). Although the majority of markers are expressed by both PBMCs and dMCs, cells from the two tissues segregate into distinct hemispheres suggesting that there is minimal overlap in phenotype. Some markers are largely restricted to a single hemisphere and known differences are confirmed by the tSNE analysis (Supplementary Fig. 4). For example, CD16 and CD57 are enriched in the peripheral blood hemisphere which contained few cells staining for tissue-residency markers, CD49a and CD103, which bind collagen IV and E-Cadherin respectively (Fig. 1d).

**Characterisation of dILC subsets.** The initial tSNE analysis shows considerable heterogeneity within the dILC niche. To characterise this diversity and identify which dILC subsets are present, a new tSNE map was generated using Lin- dILCs, incorporating the surface and intracellular markers indicated in the final column of Supplementary Table 1. Clustering by DensVM reveals 13 clusters which can be further annotated based on their profiles of marker expression (Fig. 2a, b; Table 1)[30]. Some of the clusters present correspond to dNK subsets, dNK1-3, previously identified by our scRNA-seq analysis (Supplementary Fig. 5). This CyTOF analysis permits separation of additional clusters at high resolution.

The main phenotypic characteristics of the ILC clusters we have identified are shown in Table 1. Some of these dNK and ILC subsets are not easily distinguished by simple marker combinations using traditional 2D gating strategies. The fact that they can be clearly identified using mass cytometry combined with tSNE, illustrates the utility of this approach. The four clusters 10–13 (c10–13) are all characterised by high levels of KIR expression. Together, they represent the dNK1 subset and the clusters are distinguished by their combinatorial KIR expression patterns. Cluster 9 (c9) identifies a distinct dNK subset expressing high levels of NKG2A but lower levels of KIR, and Eomes compared with dNK1. Based on similarities to the dNK2 subset identified by scRNA-seq we refer to c9 as dNK2 hereafter (Supplementary Fig. 5). Clusters 5 and 8 (c5, c8), which correspond to dNK3,

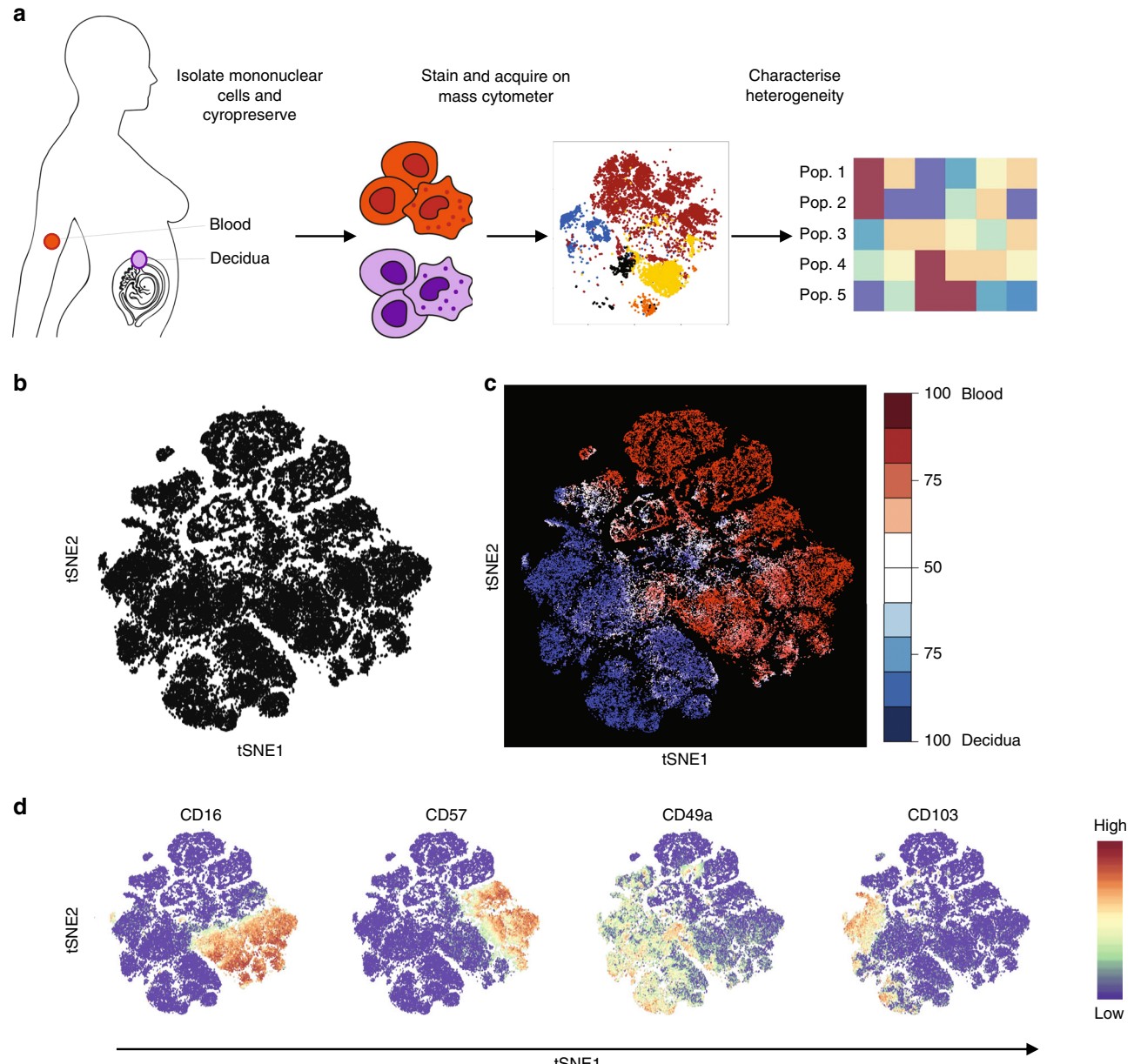

**Fig. 1 Decidual ILC subsets identified by mass cytometry. a** Workflow summary for the analysis of cryopreserved matched blood and decidual samples by mass cytometry. **b** t-Distributed Stochastic neighbour embedding (tSNE) landscape of matched CD45+ Lineage negative (Lin−) (CD3, CD19, CD14, HLA-DR) decidual and peripheral blood mononuclear cells ($n = 6$). The algorithm incorporated markers indicated in final column of Supplementary Table 1 to generate the tSNE landscape. **c** Nearest-neighbour-based tissue probability plot indicates the relative distribution of decidual and peripheral blood derived CD45+ Lin− cells within an area of the tSNE landscape. Blue = surrounded by decidual cells, Red = surrounded by peripheral blood cells, White = surrounded by mixed cells. **d** The tSNE landscape is coloured according to relative expression intensity of indicated markers. Red = high expression, Blue = low expression. Cytofkit and R studio were used to perform these analyses. Source data are provided as a Source Data file.

differ in their NKp44 and NKG2A expression and resemble ieILC1 in other mucosal sites. A proliferative subset with high Ki-67 levels corresponding to dNKp (c7) is also present. dNK are known to proliferate in vivo and based on their expression of moderate levels of KIR and NKG2A and low expression of CD117 and CD127, we believe the Ki-67+dNKp population represent a mixture of dNK cells, including dNK1, that are dividing within decidual tissue[10]. dNK1 (c10–13) are the most abundant subset identified in cryopreserved dILCs (30%), followed by dNK2 (c9, ~15%) and dNK3 (c5, c8, ~15%) (Fig. 2c). The proportion of dNK1 in freshly isolated samples was even greater than for cryopreserved samples (Supplementary Fig. 2F).

Although ILCs often show tissue-specific phenotypes, which can present problems for identification, we confirm that a decidual ILC3 cluster (dILC3) which is CD127+CD117+ is present (c6) containing both LTi-like and NCR+dILC3s[7]. We could not confidently identify dILC2s with our panel due to the absence of ILC2-specific markers, but small numbers of CD56−CD127+CD161+Tbet− cells are present within c3, suggestive of ILC2[28]. Previous authors have described three main dNK populations, defined by NKp44 and CD103 expression[6]. We find these populations spread across our tSNE map suggesting these markers do not correlate directly with the subsets defined by CyTOF (Supplementary Fig. 6).

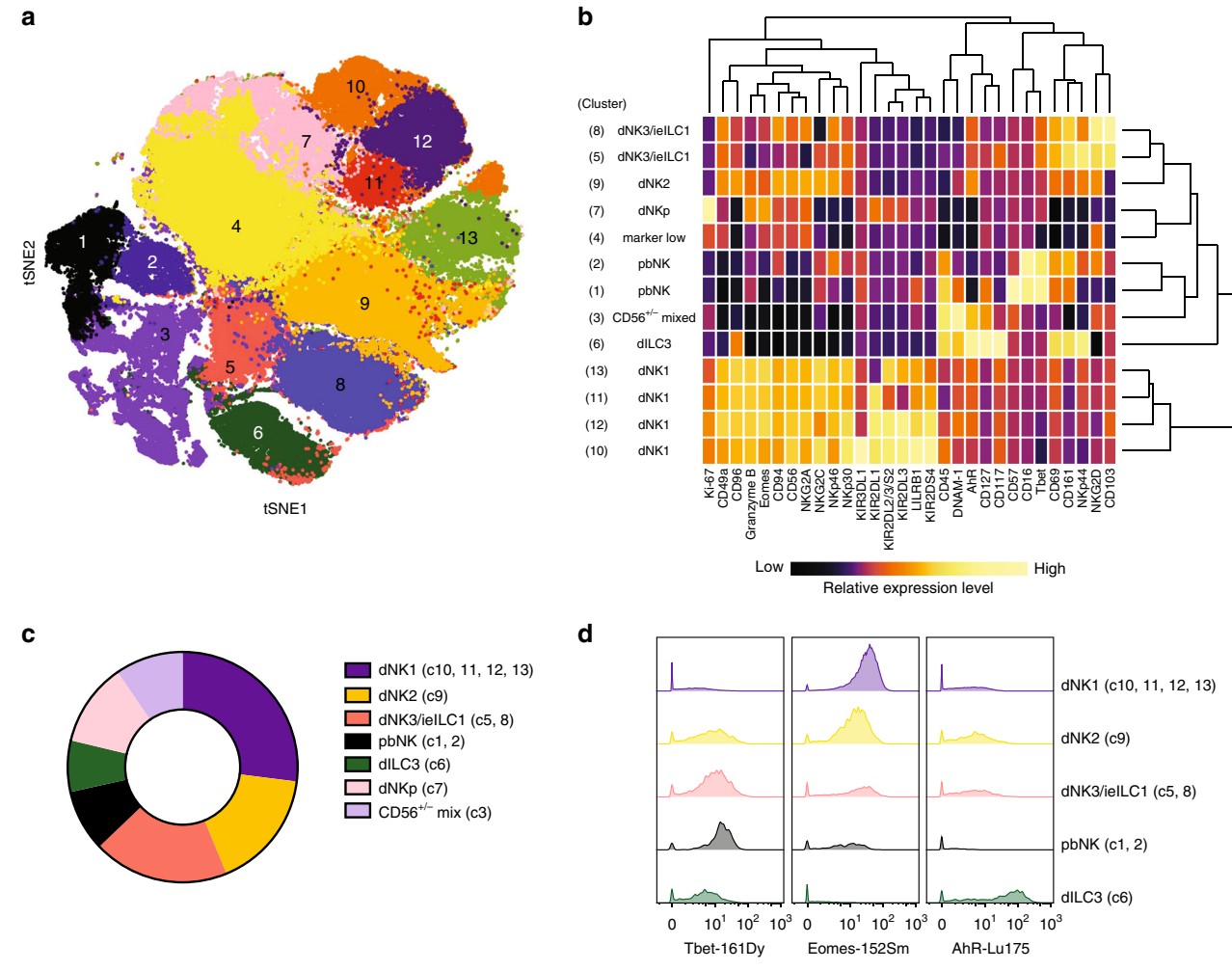

**Fig. 2 Characterisation of dILC subsets. a** Density-based local maxima cluster With SVM (DensVM) clustering of CD45+Lin− (CD3,CD19,CD14,HLA-DR) cryopreserved decidual cells using mass cytometry data on markers indicated in Supplementary Table 1 (n = 12). **b** Heatmap displaying mean expression of indicated markers across the 13 annotated clusters. Values have been scaled by column to show relative expression. **c** Donut plot showing the average frequencies of identified clusters from cryopreserved decidual cell preparations (n = 12). **d** Representative histograms for transcription factors across denoted subsets: dNK1-3, ILC3, dNKp, and pbNK. Clusters representing each subset are shown in brackets. Source data are provided as a Source Data file.

**Table 1 Identity of Lin− clusters in Fig. 2, based on phenotype.**

| Cluster | Annotation | Phenotype and comment |
|---|---|---|
| 1 | pbNK | CD49a$^{lo}$ Eomes$^{int}$ Tbet$^{hi}$ $^{(++)}$ CD16+CD57+ |
| 2 | pbNK | CD49a$^{lo}$ Eomes$^{int}$ Tbet$^{hi}$ $^{(++)}$ CD16+CD57− |
| 3 | Mixed CD56+/− | Heterogeneous group: CD56+/− CD127+/− CD161+/− CD94$^{lo}$ DNAM1+/−AhR+/− |
| 4 | Marker Low | Low expression levels for many canonical dNK markers e.g., CD56, GzmB, NKp46, NKp30. Loss of CD69, CD49a, "unhealthy" cells -increased after cryopreservation |
| 5 | dNK3 (ieILC1) | CD69$^{hi}$ Eomes$^{int}$ Tbet$^{hi}$ CD103+NKG2D$^{hi}$ CD161$^{hi}$ NKp44$^{hi}$ NKG2A$^{lo}$ |
| 6 | dILC3 | CD127$^{hi}$ CD117$^{hi}$ AhR$^{hi}$ CD94− CD56+/− NKp44+/− |
| 7 | dNKp | CD69$^{lo}$ Eomes$^{int}$ Ki-67$^{hi}$ NKG2A$^{hi}$ KIR+/− |
| 8 | dNK3 (ieILC1) | CD69$^{hi}$ Eomes$^{int}$ Tbet$^{hi}$ CD103+NKG2D$^{hi}$ CD161$^{hi}$ NKp44$^{lo}$ NKG2A$^{hi}$ |
| 9 | dNK2 | CD49a$^{int}$ Eomes$^{int}$ Tbet$^{int}$ CD103− NKG2A$^{hi}$ KIR$^{lo}$ |
| 10 | dNK1 | CD49a$^{hi}$ Eomes$^{hi}$ Tbet$^{lo}$ NKG2A$^{hi}$ LILRB1+KIR3DL1+KIR2DL1+KIR2DL3/L2/S2+ |
| 11 | dNK1 | CD49a$^{hi}$ Eomes$^{hi}$ Tbet$^{lo}$ NKG2A$^{hi}$ LILRB1+KIR3DL1− KIR2DL1+KIR2DL3/L2/S2− |
| 12 | dNK1 | CD49a$^{hi}$ Eomes$^{hi}$ Tbet$^{lo}$ NKG2A$^{hi}$ LILRB1+KIR3DL1−KIR2DL1+KIR2DL3/L2/S2+ |
| 13 | dNK1 | CD49a$^{hi}$ Eomes$^{hi}$ Tbet$^{lo}$ NKG2A$^{hi}$ LILRB1+KIR3DL1−KIR2DL1−KIR2DL3/L2/S2+ |

Decidual tissue preparations always contain peripheral blood cells and these pbNK occur as two clusters (c1, c2), distinguished by CD16 or CD57 expression. Immuno-histochemistry has only revealed sparse CD56+CD16+NK cells within the decidua itself[3]

and these clusters represent contaminants from maternal blood. Cluster 4 (c4) are CD56+ cells characterised by lower expression levels of all canonical dNK markers. These dNK have low CD49a expression, (not seen in stains of matched non-cryopreserved

cells) and are non-functional in response to phorbol-myristate acetate (PMA) plus ionomycin treatment. We conclude that they are unhealthy cells whose phenotype is altered by cryopreservation and thus they have been removed from all subsequent analyses (Supplementary Fig. 2D, E).

We also analysed the transcription factor profiles of the main dILC clusters. dNK1 (c10–c13) express the highest levels of Eomes, whilst dNK3 (c5, c8) preferentially express Tbet. In comparison, dNK2 expresses intermediate levels of both, whilst AhR is highest in dILC3 (Fig. 2d). We also looked for tissue-resident signatures, expression of inhibitory receptors for self-HLA (NKG2A, LILRB1, KIR), activating receptor profiles and granule content of identified subsets (Supplementary Fig. 5). dNK1 (c10-c13) are characterised by high levels of CD49a, LILRB1, KIR, and granzyme B. In contrast, dNK3—similar to ieILC1s—(c5, c8) express high levels of CD103, CD69, CD161, and NKG2D. dILC3 (c6) also express high levels of CD69 and CD161, but only low levels of NKG2D and granzyme B. To summarise, our findings reveal the considerable heterogeneity amongst dILC subsets with several clusters defined by phenotypic cell surface markers, intracellular transcription factors and granule proteins.

**dILC responses to unspecific stimulation or missing self**. We next measured the functional responses of dILC subsets to two different stimulants. Firstly, dMCs were treated for 4 h with PMA and ionomycin before staining with the CyTOF antibody panel (Supplementary Table 1). This included CD107a as a proxy for degranulation, and intracellular staining of cytokines and chemokines GM-CSF, XCL1, IFNγ, MIP1α (CCL3) and MIP1β (CCL4). GM-CSF (Granulocyte-macrophage colony-stimulating factor) stimulates haematopoietic stem cells to produce granulocytes and monocytes, but more relevant to the decidua is its ability to attract EVT[17]. The role of IFNγ in human decidua is unclear, whereas it is an essential cytokine for vascular remodelling in murine decidua[31]. XCL1 (lymphotactin), is believed to act on specific antigen-presenting DC subsets in tissues and the receptor is also expressed on EVT[15]. MIP1α (Macrophage inflammatory protein-1α) binds CCR1, CCR4, and CCR5 and MIP1β binds CCR5 and CCR8; in other tissues they have been shown to recruit various leucocytes, including NK cells, neutrophils and monocytes.

We generated a new tSNE map after dNK stimulation and used this to gate manually on the Lin−CD56+ subsets previously defined in Fig. 2; Table 1. For simplicity, we have grouped the clusters into conventional pbNK (c1, c2), dNK1 (c10–c13), dNK2 (c9) and dNK3 (c5, c8) (Table 1). This global analysis shows that dILC subsets differ significantly in their capacity to respond (Fig. 3a and Supplementary Table 4). Consistently across all functional readouts, the most responsive subset is dNK3, followed by dNK2. The contaminating pbNK present express CD107a and produce larger amounts of IFNγ. The most abundant cells, dNK1, are functionally muted in comparison with dNK3 or dNK2 and they preferentially produce chemokines such as MIP1β (Supplementary Fig. 7 and Supplementary Table 4). NCR+dILC3s produce GM-CSF and XCL1 but little IFNγ[6,32]. The proliferating cells, dNKp, show low responses for all readouts. The ability of each dNK subset to mount a polyfunctional response to stimulation was assessed by examining three functional readouts: CD107a staining or IFNγ or at least one of the cytokines MIP1α, MIP1β, GM-CSF, XCL1 (Fig. 3b). Within responding cells, dNK3 showed the strongest polyfunctional responses; more than 75% upregulating at least two readouts simultaneously, with ~37% of cells expressing all three. In contrast, 60% of dNK1 expressed a single readout, most of which were CD107a− IFNγ−, indicating these cells are specialised for secretion of cytokines

other than IFNγ. dNK2 displayed intermediate responses. Thus all three dNK subsets show distinctive patterns of polyfunctional responses compared to pbNK cells.

Secondly, we co-cultured dMCs with the HLA-deficient, K562 cell line in a classical missing-self assay. The majority of dNK cells express NKG2A and are educated to detect missing self. Moreover we selected donors who were heterozygous for group 1 and group 2 HLA-C alleles (C1/C2) so that both KIR2DL1+ and KIR2DL3 +cells are further educated[33,34]. As with PMA/ionomycin, dNK3 and dNK2 are the most responsive to missing-self (Supplementary Fig. 8 and Supplementary Table 5), but in this assay dNK2 responses are higher than dNK3. The relative magnitude of response of each subset therefore depends on the specific stimulus. dNK1 cells are once again low responders, consistent with our previous findings that show decreased degranulation of dNK subsets expressing multiple inhibitory NKR in response to K562[35]. To determine if there is a more functional subset within the dNK1 compartment, we focused on CD103 because, although this is a marker of ieILC1s (and indeed is expressed by dNK3), we also see expression on ~30% of dNK1. CD103+dNK1 produce significantly higher levels of CD107a, MIP1α, MIP1β, and XCL1 in response to PMA/ionomycin, and increased levels of MIP1β and XCL1 upon co-culture with K562 cells when compared to CD103− dNK1 (Fig. 3c, d). To summarise, functional differences exist within the dILC compartment; in both assays dNK3 and dNK2 subsets show the highest responses, but which subset predominates is stimulus dependent. We find that particular cytokines/chemokines are produced by specific subsets and that CD103 expression by some dNK1 correlates with increased functional responses, demonstrating further heterogeneity within this subset.

**KIR acquisition on dNK and pbNK**. Despite the weak functional responses to PMA/ionomycin and missing self, the major dILC subset, dNK1, has the potential to respond to invading EVT as they express NKR, including KIR, with ligands on EVT. Immunogenetic and in vitro studies also suggest that binding of KIR on dNK to HLA-C on EVT regulates trophoblast invasion and pregnancy outcome[22,23,35,36]. In pbNK, due to the stochastic nature of KIR expression[37], individual NK cells can express multiple KIR and acquisition of KIR, CD57 and the loss of NKG2A is associated with NK cell differentiation[38]. Importantly, acquisition of inhibitory KIR that bind self HLA class I molecules leads to more responsive pbNK cells through the process of NK cell education[33]. To study the effects of KIR on phenotype and function of dNK, we analysed how additional NKR, transcription factors and proteins present in NK granules correlate with KIR acquisition on Lin− CD56+dMCs. We confirm the findings on pbNK that CD57 and CD16 expression increases with acquisition of KIR and find DNAM-1 also increases even when KIRs are not stratified by education status (Fig. 4a)[39]. LILRB1, Ki-67, NKp30, and granzyme B, all increase markedly on dNK with increasing KIR co-expression; these markers are either constant or show modest changes on pbNK (Fig. 4b). NKG2A expression decreases on pbNK with increasing KIR co-expression[38] but the majority of dNK are NKG2A+ and this slightly increases with KIR acquisition (Fig. 4c). In contrast, NKG2D, CD161, and Tbet decrease on dNK as KIR co-expression increases; these markers are maintained on KIR+pbNK (Fig. 4d). Therefore, acquisition of KIR in dNK is associated with very different changes in key phenotypic markers compared to pbNK. These include LILRB1, NKG2A, the transcription factor Tbet and granzyme B.

**dNK granule content and organisation**. We have previously shown that KIR+ dNK express higher levels of perforin,

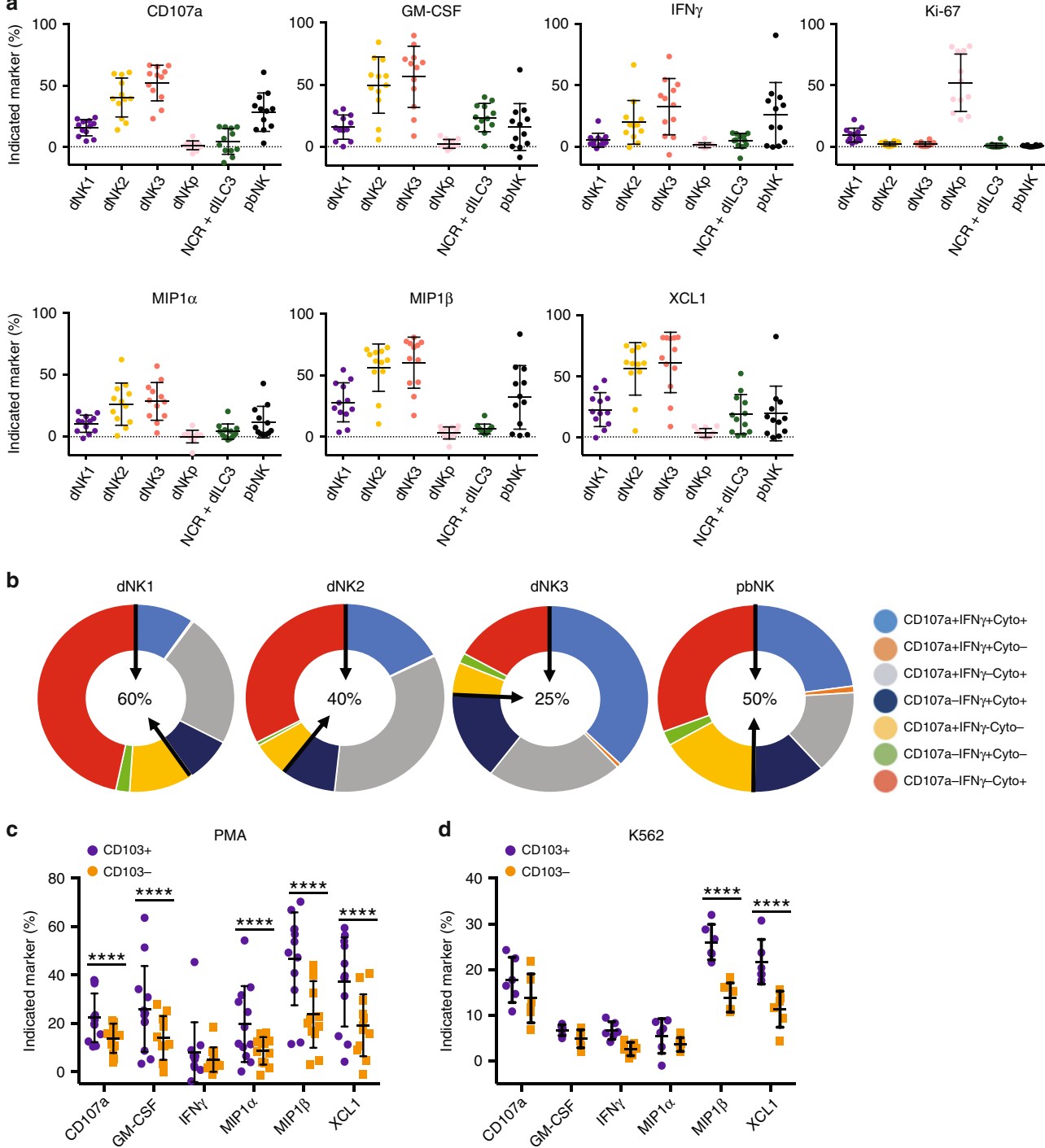

**Fig. 3 dILC responses to unspecific stimulation or missing self.** a Decidual mononuclear cells were stimulated for 4 h by PMA plus ionomycin in the presence of protein transport inhibitors. A new tSNE landscape was generated using the markers indicated in the final column of Supplementary Table 1. Lineage negative (Lin−) CD56+ subsets identified in Fig. 2 were manually gated on the tSNE plot to show dNK1 (c10–c13), dNK2 (c9), dNK3 (c5, c8), dILC3 (c6), dNKp (c7) and pbNK (c1, c2). Scatter plots show proportion of cells within each subset staining for functional readouts or Ki-67 (n = 12). Values represent stimulation minus unstimulated media control for each subset. Error bars represent SD. **b** Pie charts show the mean percentage of each subset that express one, two or three functional readouts as a measure of polyfunctionality, following stimulation. Readouts were: CD107a and/or IFNγ and/or at least one of the cytokines MIP1α, MIP1β, GM-CSF, XCL1 (Cyto). Non-responding cells that expressed none of the functional readouts were excluded. Figure in centre shows percentage of responding cells that express only one of the readouts (red, green and yellow segments, delineated by black arrows) (dNK1–3 n = 12, pbNK n = 7). **c** dNK1 are stratified by CD103 expression in response to PMA plus ionomycin (n = 12) and **d** 6 h co-culture with K562 (n = 6). Error bars represent SD. **a** Two-tailed one-way ANOVA of matched data points with Tukey's correction was performed. **c, d** A RM two-way ANOVA of matched data points was performed with Sidak's multiple comparisons test. p-values of comparisons for **a** are found in Supplementary Table 4. Where indicated, ****p < 0.0001. Source data are provided as a Source Data file.

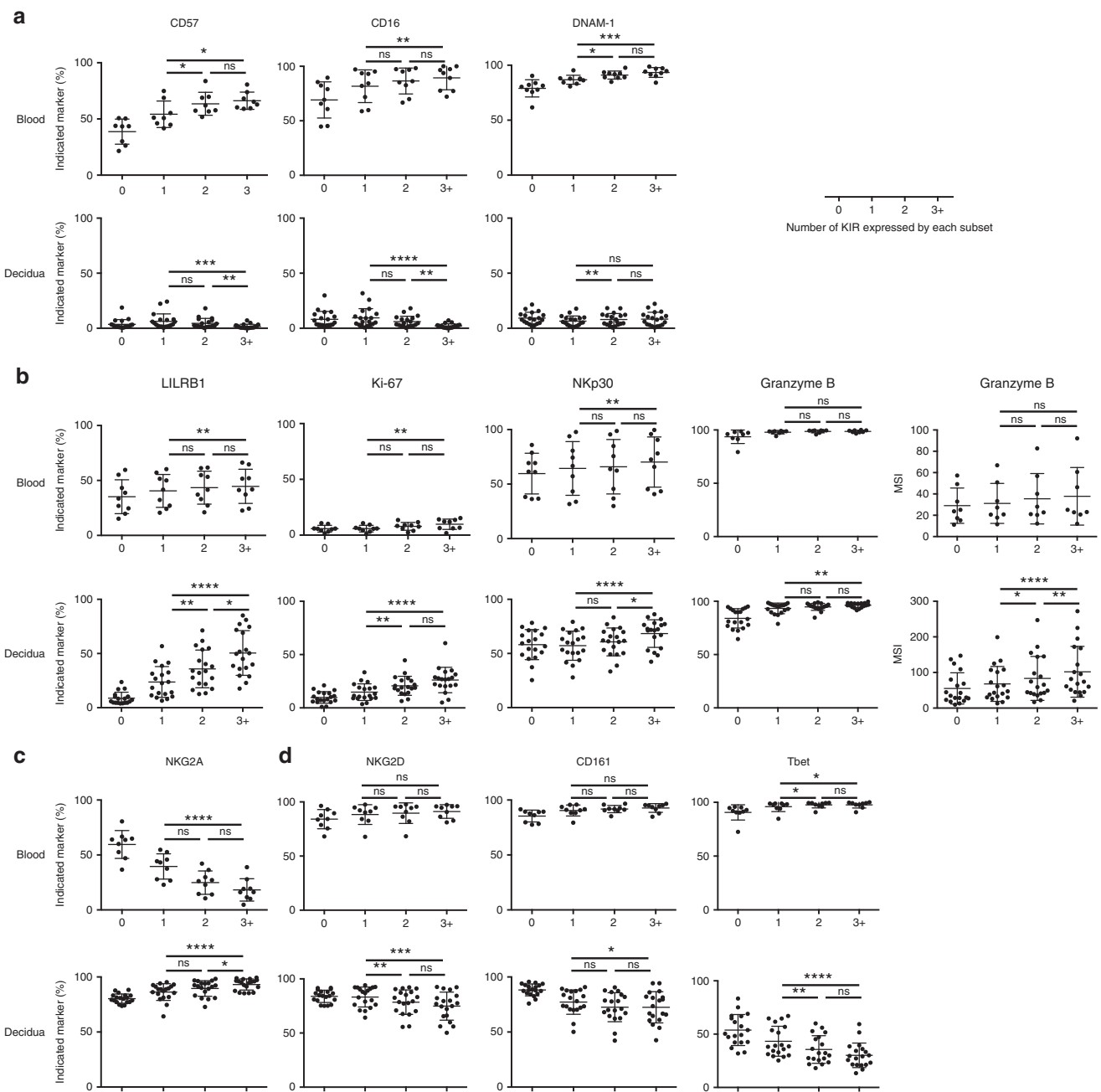

**Fig. 4 KIR acquisition on dNK and pbNK.** Boolean combinations of KIR receptors were gated manually on Lineage negative (Lin−) CD56+dMCs and Lin− CD56dim PBMCs following mass cytometry and grouped according to the number of KIR expressed from zero to three or more KIR (3+). Markers are grouped in the following way **a** increase on CD56dim pbNK, **b** increase on dNK, **c** decrease on CD56dim pbNK, **d** decrease on dNK (n = 19 decidua and 8–9 peripheral blood, five pregnant and four non-pregnant blood donors). Friedman tests using Dunn's correction for multiple testing were performed for KIR = 1, 2, 3+. Error bars represent SD. *p < 0.05, **p < 0.01, ***p < 0.001, ****p < 0001. Source data are provided as a Source Data file.

granulysin and granzymes A and B than KIR− dNK[8]. We further explored this connection between granule content and KIR expression on Lin− CD56+dMCs. We find that as dNK acquire more KIR the flow cytometry side-scatter (a proxy for granularity) increases with each additional KIR. This holds true for dNK expressing either single inhibitory or activating KIR (Fig. 5a–d). In contrast, forward-scatter (a proxy for cell size) did not increase (Supplementary Fig. 9A, B). Using flow cytometry, we went on to analyse the quantities of perforin, granzyme B and granulysin in dNK, compared to pbNK subsets (Supplementary Fig. 10A). We confirm previous findings that perforin and granzyme B levels in dNK are intermediate

between CD56[bright] and CD56[dim] pbNK, whereas dNK have higher levels of granulysin compared to both pbNK subsets[40] (Supplementary Fig. 10B).

To investigate the morphology and location of the granules in more detail, we used electron and confocal microscopy of dNK and pbNK. Granules in dNK are significantly larger than those in pbNK (~600 nm diameter for dNK, ~200 nm for pbNK) (Fig. 5e, f) and fewer in number[41]. Confocal microscopy shows that perforin, granzymes and granulysin all localised to fewer, larger structures, consistent with localisation in enlarged granules as seen by EM. (Fig. 5g). Few large granules are characteristic of NK cells from CHS patients that are also poor killers[27]. CHS NK

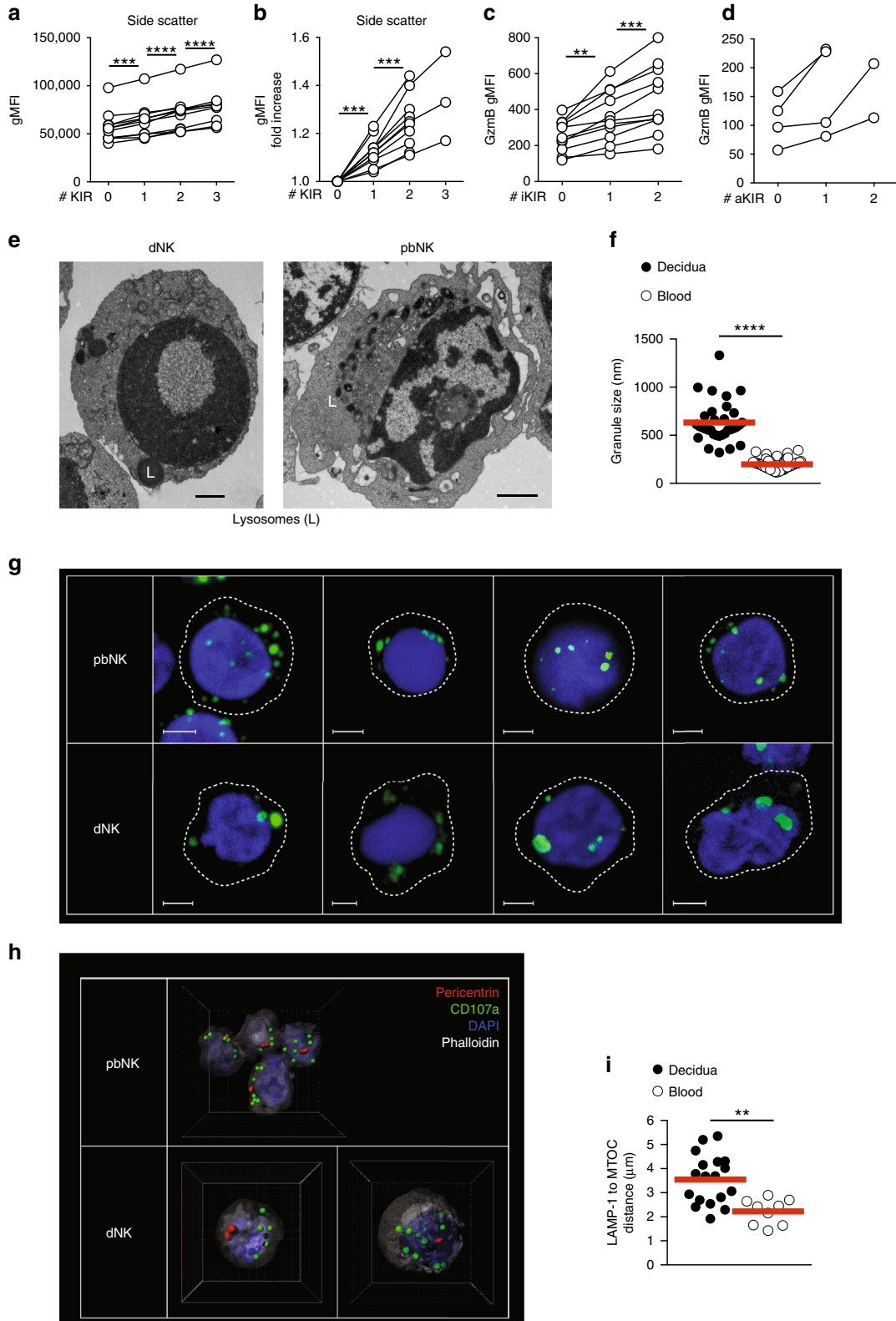

cell granules are less clustered around the MTOC and we therefore measured this in dNK[25,27]. Analysis of dNK by confocal microscopy stained for LAMP-1/CD107a and pericentrin (an MTOC-associated protein) shows that dNK granules are further away from the MTOC compared to pbNK (Fig. 5h, i)[42]. In summary, dNK have distinctive organisation and distribution of granules compared with normal pbNK. In contrast to pbNK, the content of granule proteins such as granzyme B increases as dNK acquire KIR.

**dNK responses triggered by activating KIR.** pbNK granules function as $Ca^{2+}$ ion stores that contribute to increased signalling capacity so that cells with larger granules produce more

**Fig. 5 dNK granule content and organisation. a** Granule content represented as geometric mean fluorescent intensity (gMFI) of side scatter in Lineage negative (Lin−) CD56+ cell subsets co-expressing 0–3 Killer-cell Immunoglobulin-like Receptors (KIRs, $n = 11$). Lines connect dNK subsets from the same individual. **b** Fold increase in side scatter over previous data-point in **a**. **c** Geometrical mean fluorescence intensity (gMFI) of Granzyme B in KIR2DS1-KIR2DS4- Lin-CD56+ cells co-expressing 0–2 inhibitory KIRs ($n = 11$). **d** Granzyme B gMFI in KIR2DL1-KIR2DL2/S2/3− Lin-CD56+ cells co-expressing 0–2 activating KIRs ($n = 4$). **e** Live Lin-CD56+CD94+dNK and pbNK were FACS-sorted and visualised using electron microscopy. Electron dense lysosomal structures are indicated with "L". Data is representative of four donors. Scale bar indicates 1 µm. **f** Granule diameter measured by electron microscopy, aggregated data for multiple cells ($n = 33$ for dNK and 53 for pbNK). Red lines indicate mean. **g** Confocal microscopy pictures of pbNK (top) and dNK bottom) stained with the markers indicated (green), nucleus (blue) and cell outline (dashed line). Representative 2D views of multiple cells from five experiments are shown. White scale bar indicates 2 µm. **h** 3D representation of four pbNK (top) and two dNK cells (bottom) stained for centrosome (MTOC, pericentrin, red), CD107a/LAMP-1 (green) and nucleus (blue). Spots (CD107a/LAMP-1) and surfaces (pericentrin, DAPI for nucleus and Phalloidin for cell surface) created in Imaris are shown. The side of each square in the 3D cube represents 2 µm. **i** Quantification of multiple distance measurements from CD107a/LAMP-1 structure centre (green spots) to centrosome centre (red surfaces) in **h**. Line indicates mean. Dots represent average distances in each cell analysed ($n = 18$ and 9), from three different experiments with unique unmatched donors. **a–c** Stars indicate significance obtained with two-tailed one-way ANOVA of matched data points with Tukey correction. Tests were only performed on full series. **$p < 0.01$, ***$p < 0.001$, ****$p < 0.0001$. **f, i** Stars indicate significance obtained with a two-tailed Mann–Whitney test with 95% confidence level, ****$p < 0.0001$, **$p < 0.001$. Source data are provided as a Source Data file.

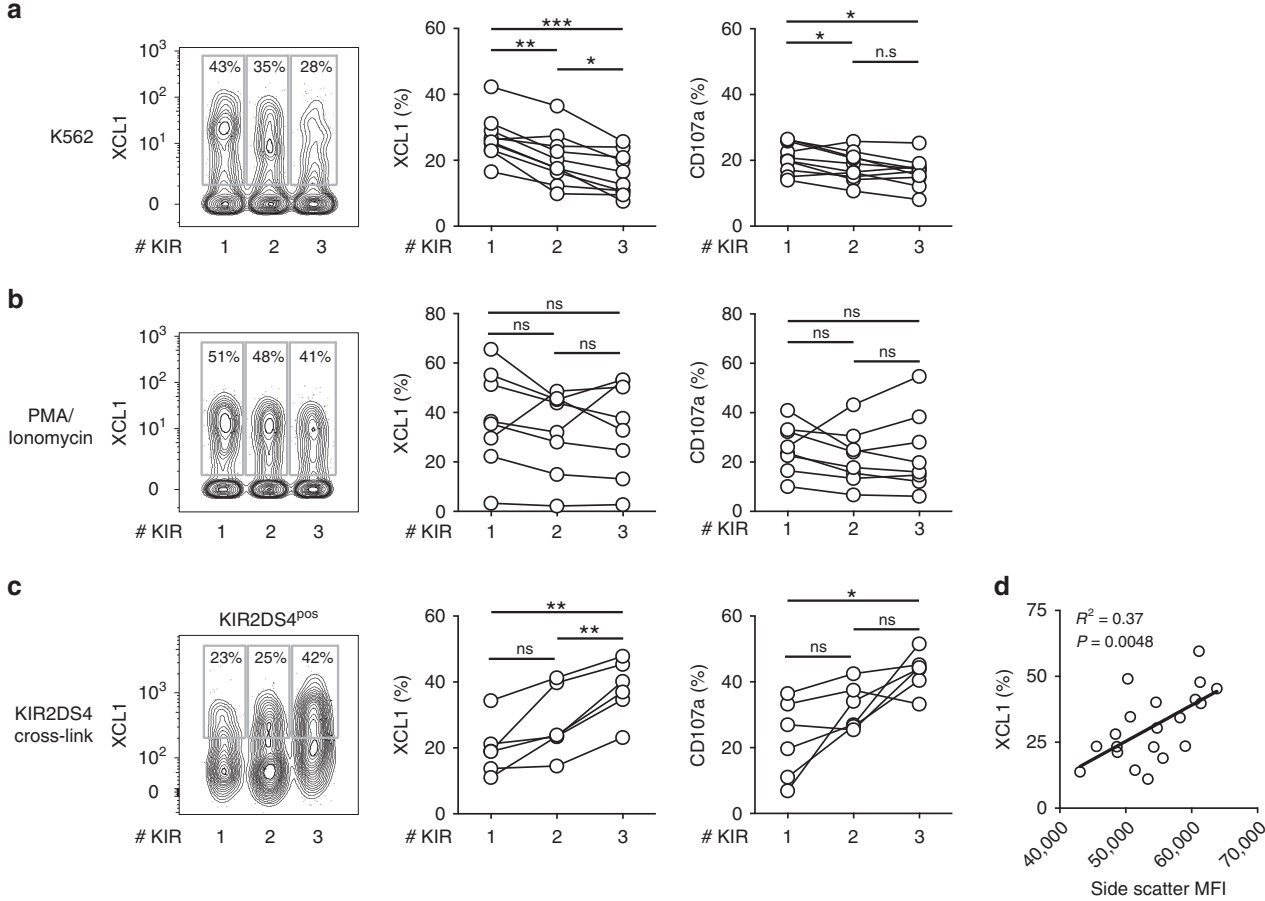

**Fig. 6 dNK responses triggered by activating KIR. a** Representative XCL1 and CD107a staining by CyTOF in lineage negative (Lin−) CD56+ decidual cells co-expressing 1–3 Killer-cell immunoglobulin-like receptors (KIR) following a 6 h co-culture with K562 ($n = 10$). **b** Representative XCL1 and CD107a staining by mass cytometry in Lin-CD56+ cells co-expressing 1–3 KIR following 4 h stimulation by PMA plus ionomycin ($n = 8$). **c** Representative XCL1 staining by flow cytometry in Lin-CD56+KIR2DS4+ decidual cells co-expressing 1–3 additional KIRs following activation via P815 cells coated with anti-KIR2DS4 ($n = 6$). **d** Correlation of frequency of XCL1+Lin-CD56+ decidual cells and mean side scatter for the same subset ($n = 20$). Two-tailed p-value calculated for Pearson correlation coefficients. Two-tailed one-way ANOVA of matched data points with Tukey correction and 95% confidence level was used. *$p < 0.05$, **$p < 0.01$, ***$p < 0.001$. Source data are provided as a Source Data file.

cytokines when an activating receptor is triggered[24]. We next asked whether Lin− CD56+dMCs become more functional as granule content increases with acquisition of KIR (Figs. 4b, 5a–d). When analysed by FACS, these Lin− CD56+KIR+

correspond mostly to dNK1 identified by mass cytometry. We confirm that, unlike pbNK, degranulation of dNK decreases with increasing KIR expression in response to missing self, and cytokine production follows this trend (Fig. 6a)[43]. In contrast,

following PMA/ionomycin stimulation, there is no obvious decrease in response with increasing KIR expression (Fig. 6b). However, these are not physiological stimuli. We have shown that triggering activating KIR2DS1[13,17] and KIR2DS4[15] on dNK to mimic recognition of HLA-C on trophoblast, results in production of cytokines that enhance EVT cell migration in vitro. Furthermore, these activating KIR are associated with a reduced risk of developing pre-eclampsia[15,22,35,36]. When dNK are stimulated by cross-linking KIR2DS4, we now show that increased KIR co-expression on KIR2DS4+dNK is associated with increased cytokine production and degranulation (Fig. 6c). A higher proportion of dNK express both readouts as the number of KIR expressed goes up, so polyfunctionality of dNK correlates with KIR co-expression. dNK with highest side scatter also have the highest cytokine production supporting the link with increased granule content (Fig. 6d). In summary, dNK with more KIR display increased functionality when triggered via an activating KIR in contrast to stimulation via missing self, where dNK responsiveness decreases with more KIR.

## Discussion

We have used CyTOF to profile the phenotype and function of tissue-resident ILCs in human decidua. Comparison of 13 phenotypic subsets of Lin− dILCs by CyTOF with scRNAseq transcriptomic profiles defining dNK1–3, dNKp, dILC3, and pbNK confirms the identity of the major blood and decidual ILC populations at the protein level with a greater degree of resolution. Four clusters of dNK1 are defined by KIR co-expression patterns and two dNK3 sub-populations are distinguished by NKp44 and NKG2A expression. Based on the similarities between our RNA profiles and CyTOF data we also confirm the presence of the dNK2 subset. NK transcription factors are underrepresented in scRNAseq due to their low expression levels; here we find that dNK1 are Eomes$^{high}$Tbet$^{low}$, in contrast to dNK3 which are Eomes$^{int}$Tbet$^{high}$. dNK2 express intermediate levels of both Eomes and Tbet suggesting that they might be positioned along a differentiation trajectory between dNK3 and dNK1. In agreement with previous reports, NCR+ILC3 and LTi-like dILC3s are present but it is difficult to reliably define the sparse dILC2 subset[4,5,7].

Heterogeneity of dILC has been described by conventional flow cytometry for a range of phenotypic markers including KIR, other NKR and integrins[6,44,45]. For example, three dNK populations were defined by flow cytometry on the basis of NKp44 and CD103 expression but comparison with our high dimensional analysis using CyTOF (Supplementary Fig. 6) shows the difficulty of capturing the true complexity of dILCs using only a few markers[6]. We also did not identify a cluster directly corresponding to "pregnancy trained dNK cells" (PTdNK), although they do most closely resemble dNK1 that have higher levels of NKG2C, LILRB1, KIR, and glycolytic enzymes[8]. Decidual PTdNK proportionally increase in women after their first pregnancy[12] but this clinical information was not available under our ethical permission for the study.

We obtained this data using frozen cells but there are two caveats: some markers such as CD49a are more sensitive to the freeze/thawing process and a cluster of unhealthy cells was removed from the analysis because it is largely absent from freshly isolated cells. This is a small study and we were therefore unable to detect differences resulting from gestational age. Characterising the spatial dynamics of dILC subsets was also not within the scope of this study. Finally, as the samples are from early in gestation, how dILCs are altered in disorders of placentation such as pre-eclampsia which only present later in pregnancy is undetermined.

The relationship of these human dILC subsets to those in murine decidua is unclear. There are three murine Group 1 ILC subsets: CD49a+Eomes- uterine ILC1 (uILC1), CD49a+Eomes+trNK and CD49a-Eomes+conventional NK (cNK)[4,46,47]. Murine uILC1 express CXCR6 and may represent memory cells as they expand in second pregnancies[46]. They therefore could be functionally analogous to KIR+PTdNK, although they have low levels of Ly49. trNK resemble dNK1 with higher levels of Ly49, Eomes and Ki-67[46,47]. Although murine cNK are similar to our pbNK clusters, these are abundant within murine decidua basalis whilst CD16+pbNK are only sparse in human decidual tissue[4,48]. Human dNK3 resemble ieILC1s for which a murine counterpart has been characterised in the gut (NKp46+NK1.1+CD160+)[49]. However, uterine trNK and uILC1 are both present in Nfil3$^{-/-}$ mice, so neither subset directly corresponds to NFIL3-dependent gut ieILC1s[50]. The inability to easily correlate murine and human uterine ILC subsets could reflect the considerable anatomical differences in placentation between mice and humans. A better functional characterisation of dILC subsets in both species may reveal functional homologies among phenotypically different cells.

dNK are phenotypically and functionally unlike other trNK present in many human tissues[9]. CD49a+liver-resident NK cells (lrNK) express KIR but not NKG2A, whilst CXCR6+lrNK express NKG2A and not KIR[51–53]. The main lung NK cells are circulating CD56$^{dim}$ CD16+, with a smaller CD56$^{bright}$ NK population expressing CD69, CD49a, and CD103[54,55]. Unlike dNK1, these CD56$^{bright}$ lung NK express less KIR2DL2/L3 than lung CD56$^{dim}$CD16+NK[54]. Differentiating CD56$^{dim}$ pbNK acquire KIR and CD57, lose NKG2A, and increase responsiveness with acquisition of inhibitory KIR specific for self-MHC, through NK education[33,38]. dNK are quite different because as KIR co-expression increases, we find dNK1 exhibit decreased responsiveness to stimulation by missing self, but greater responses to cross-linking activating KIR2DS4. This paradoxical finding might be explained by our findings that side-scatter and granzyme B expression also rise with increasing KIR, suggesting changes in granule content and organisation[8,40]. We also find that the increased levels of granzyme B reported in dNK expressing KIR2DS1+[40], occurs with both activating and inhibitory KIR.

The different functional responses of dNK and pbNK as they acquire more KIR, may be due to the differences observed in granule organisation between the two. Granzyme B accumulates in granules corresponding to secretory lysosomes and here we show that dNK granules are larger and located further away from the MTOC compared to resting pbNK. dNK were previously shown to be unable to polarise their MTOCs and perforin-containing granules to the immune synapse[56]. Enlarged granules and higher granzyme B expression are linked to increased functional capability in pbNK[24]. In pbNK, larger granules appear to act as stores leading to increased Ca$^{2+}$ release upon receptor cross-linking and greater degranulation and cytokine release. The parallel increase in granule proteins and responsiveness to KIR cross linking as number of KIR increases, suggests a similar mechanism may operate in dNK. All these features of dNK granules resemble the pbNK from CHS patients that are poorly cytotoxic but maintain the capacity to produce cytokines[25,26,42]. The genetic mutation responsible for CHS affects the lysosomal trafficking regulator, LYST. Lyst is mutated in beige mice who reproduce normally and show similar morphological and functional defects to CHS patients in peripheral but not in uterine NK cells[57,58]. Furthermore, normal pregnancy is reported in CHS patients[59]. Although a reliable antibody is not available, LYST mRNA levels are lower in dNK compared to CD56$^{dim}$ pbNK[8,60]. Future work is needed to study the biology of these unusual dNK granules. Indeed, the presence of unique cells in decidua with

large cytoplasmic granules, led to the original discovery of uterine NK cells. Their large granules have unique tinctorial properties (phloxine tartrazine in humans and the lectin DBA in mice) not seen in NK cells in other tissues[61,62].

The major dILC subsets (dNK1-3, dILC3) produce factors (GM-CSF, XCL1, MIP1α, and MIP1β) whose receptors are expressed by EVT and thus are likely to modify invasion. This is stimulus dependent and does not always correlate with the resting mRNA levels found from scRNAseq[8]. Indeed, the dominant cells, dNK1, whose receptor profile suggests direct interactions with EVT, respond poorly to classical methods used to stimulate NK. Instead, when trophoblast recognition is simulated by cross-linking of KIR2DS4, these cells degranulate and make XCL1. KIR and their HLA-C ligands are highly polymorphic and immuno-genetic studies show that specific combinations of maternal KIR and their HLA-C ligands leading to dNK inhibition are associated with fetal growth restriction and pre-eclampsia where trophoblast transformation of uterine arteries is defective. Combinations that promote dNK activation are associated with enhanced fetal growth[22,23,36]. Our results show how KIR may regulate responses of the dNK1 subset to EVT during pregnancy. Increasing KIR expression in dNK1 is associated with higher granzyme B levels, changes in the organisation of granules and with higher responsiveness of dNK1 to cross-linking activating KIR2DS4 but lower responses in missing self assays. This suggests responsiveness is regulated differently by KIR in dNK compared to pbNK.

The exact roles of the other dILC subsets remains less clear. dNK2 and dNK3 produce significantly higher levels of cytokines such as XCL1, than dNK1 upon missing-self or PMA/ionomycin stimulation. The only maternal cell expressing XCR1 in decidua are dendritic cells (cDC1)[8] indicating XCL1 derived from dNK2 and dNK3 may recruit and modulate the functions of cDC1 in decidua as demonstrated in the tumour microenvironment[63]. dNK2 and dNK3 express NKR for non-HLA ligands expressed on EVT—TIGIT binds to PVR and KLRB1 binding to CLEC2D. They also secrete MIP1α (CCL3) and MIP1β (CCL4) which can bind to CCR1 and the scavenging receptor, ACKR2 (D6), both expressed by EVT[64,65]. Loss of Ackr2 in murine fetal cells leads to placental defects and fetal death[66]. To investigate the effects of cytokines and chemokines derived from dILC subsets in humans requires in vitro experimental models. We have now developed 3D organoid culture systems derived from both maternal endometria and placentas[8,67]. The trophoblast organoids can be induced to differentiate into invasive HLA-G+EVT. This characterisation of the unique features of decidual ILCs, will now permit future studies of their functional interactions with other decidual cells as well as fetal trophoblast. The uterine mucosa is dedicated to supporting the developing placenta and these distinctive ILCs must be pivotal in ensuring reproductive success.

## Methods

**Patient samples**. Decidual samples were obtained from healthy women with apparently normal pregnancies undergoing elective first trimester terminations (7–12 weeks) ($n = 28$). The age and parity of the women is not known to us. All decidua donors supplied fully informed consent and ethical approval was granted by the Cambridge Research Ethics Committee (study 04/Q0108/23). Peripheral blood was taken from women undergoing elective first trimester terminations (7–12 weeks). In addition, whole blood samples were purchased from the NHS Blood and Transplant unit ($n = 4$, used in Fig. 4). The age of donors and sex of NHS Blood and Transplant donors is unknown. All peripheral blood donors supplied fully informed consent and ethical approval was granted by the Cambridge Research Ethics Committee (study 04/Q0108/23).

**Cell lines**. The murine cell line P815 and human cell line K562 were purchased from DSMZ in Germany. Cells were not tested for mycoplasma contamination nor authenticated and were cultured in complete medium (RPMI1640 medium, antibiotics, 10% FCS).

**Isolating mononuclear cells from decidua and whole blood**. Matched peripheral blood and decidua samples were collected from women undergoing elective terminations of first trimester pregnancies. Peripheral blood mononuclear cells (PBMCs) were isolated from whole blood by Pancoll (PAN-Biotech) and cryo-preserved in 90%FCS/10%DMSO. To isolate decidual mononuclear cells (dMCs), decidual tissue pieces were first washed in RPMI-1640 and remaining blood clots and vessels removed using scalpels. Remaining tissue was then minced and 5 ml of Collagenase IV (0.1 g/100 ml, sigma) in RPMI-1640 with 10% FCS added for further dissociation by GentleMACS. The tissue was incubated for another 45 min at 37 °C whilst being gently shaken. The collagenase was quenched by addition of RPMI-1640 and then filtered through 100 μm and then 40 μm filters. dMCs were isolated following a Pancoll centrifugation step and then cryopreserved in 90%FCS/10%DMSO. This protocol has been adapted from Male et al. 2012[68]. Additional whole blood samples were purchased from the NHS Blood and Transplant unit and PBMCs were isolated and cryopreserved as described above.

**Cell staining and data acquisition by mass cytometry**. PBMCs and dMCs were thawed and washed in complete medium. Cells were then rested overnight at 37 °C with 5% CO₂ in complete medium supplemented with 2.5 ng/ml IL-15 (Peprotech). After resting, cells were washed in serum free RPMI 1640 medium and a viability stain performed using 500 μM rhodium DNA intercalator diluted 1 in 500 (Fluidigm Sciences) for 15 min at 37 °C with 5% CO₂. Cells were washed in cell staining medium (CSM, 1× PBS, 0.03% BSA, 2 mM EDTA), counted and distributed at 2–4 × 10⁶ cells per well and stained for 1 h at 4 °C in 65 μl of filtered antibody mixture. Cells were then washed twice in CSM and fixed in 1% PFA overnight at 4 °C. After fixation, cells were washed in CSM and resuspended in Permeabilization Buffer (Invitrogen) for 45 min at 4 °C before staining with 65 μl of filtered antibody mixture for 1 h at 4 °C. Cells were then washed thrice in CSM and DNA stained using 125 μM Cell-ID Intercalator Ir (Fluidigm Sciences) diluted 1 in 10,000 in 1% PFA overnight. Finally, cells were washed twice in CSM and then twice in MiliQ water before acquisition on a Helios mass cytometer.

**Cell staining and data acquisition by flow cytometry**. PBMCs and dMCs were thawed and washed in complete medium. Cells were then rested overnight at 37 °C with 5% CO₂ in complete medium supplemented with 2.5 ng/ml IL-15 (Peprotech). Cells were then counted and distributed at approximately 1 × 10⁶ cells/well in 200 μl FACS wash (FW, 1× PBS, 2% FCS, 2 mM EDTA). Cells were stained for 1 h at 4 °C and KIR2DS1 was detected by direct addition of 11PB6 antibody to wells for the last 15 min of staining. Viability was assessed using LIVE/DEAD Aqua incubated for 30 min at 4 °C. Cells were then fixed in 1% paraformaldehyde for 10 min at room temp and later analysed on a LSR Fortessa (BD Biosciences). For intracellular staining FoxP3 transcription factor staining kit was used for fixation and subsequent intracellular staining (Invitrogen).

NK were gated on as live CD3-CD14-CD19-CD56+lymphocytes. Further gating on subsets was done as indicated in figure legends. The following antibodies and reagents were used: LIVE/DEAD discriminator (LifeTechnologies), CD3 (UCHT-1, BioLegend), CD14 (M5E2, Biolegend), CD19 (HIB19, Biolegend), CD56 (HCD56, BioLegend) NKG2A (Z199, Beckman Coulter and REA110, Miltenyi), KIR2DL1 (REA284, Miltenyi), KIR2DL1/S1 (11PB6, Miltenyi), KIR2DL3 (180701, RnD Systems), KIR2DL2/3/S2 (CH-L, BD Biosciences and GL183, Beckman Coulter), KIR2DS4 (REA860, Miltenyi), Perforin (dg9, BioLegend), Granzyme A (CB9, eBioscience), Granzyme B (GB11, BioLegend), Granulysin (9/15 kDa, DH2, BioLegend). XCL-1 (109001, RnD systems) was conjugated to fluorescent molecules using Lightning-Link (Innova Biosciences, UK). Biotinylated antibodies were detected using Streptavidin Qdot605 (LifeTechnologies).

**Antibody conjugation for mass cytometry**. Antibodies and isotopes were obtained from the sources specified in Supplementary Table 1. Where antibodies required conjugation, this was performed using MaxPar X8 labelling kits (Fluidigm Sciences) according to the manufacturer's instructions.

**Stimulation with PMA/ionomycin and missing self**. PBMCs and dMCs were thawed and rested overnight in low dose IL-15 as described above. For stimulation by PMA/Ionomycin, 1 × 10⁶ mononuclear cells were treated with eBioscience™ cell stimulation cocktail (Invitrogen) for 4 h in 96-well U bottom plates at 37 °C, 5% C0₂. Golgi Plug (BD Biosciences) and Golgi Stop (BD Biosciences) were added for the final 3 h of stimulation. For K562 stimulations, mononuclear cells were co-cultured with K562 target cells at a ratio of 10:1 for 6 h in 96-well U bottom plates at 37 °C, 5% CO₂. Golgi Plug (BD Biosciences) and Golgi Stop (BD Biosciences) were added for the final 5 h of stimulation. Cells were then stained and acquired as outlined above. To analyse polyfunctional NK responses to stimulation, Lin−CD56+ subsets were manually gated on the tSNE plot to show dNK1 (c10–c13), dNK2 (c9), dNK3 (c5, c8), and pbNK (c1, c2) subsets. Boolean gating arrays were created using FlowJo to determine the frequency of cells within these subsets that stained for one, two, or three readouts. Readouts were defined as: CD107a and/or IFNγ and/or at least one of the cytokines MIP1α, MIP1β, GM-CSF, XCL1 (Cyto). Non-responding cells that expressed none of the functional readouts were excluded.

**Stimulation with activating KIR**. Cryopreserved decidua cells from previously KIR-phenotyped KIR2DS4+ donors were used to perform redirected antibody-stimulation assays. Adherent cells were depleted by plating thawed decidua cells in T175 flasks over-night. Ten million cells per flask were used in 10 ml of RPMI 1640 with 10% heat-inactivated fetal calf serum and antibiotics, supplemented with 2.5 ng/ml IL-15 (Peprotech). Next day non-adherent cells were washed off using complete medium and cells were resuspended at $2.5 \times 10^6$/ml. P815 cells were resuspended at $0.5 \times 10^6$/ml and coated with anti-KIR2DS4 (clone 179315, R&D Sytems) or isotype control by adding 5 µg/ml antibody and incubating 30 min at four degrees.

Effector and target cells were then mixed at 5:1 ratio (100 µl of each) in a U-bottom plate. Plates were centrifuged for 2 min 300 rpm followed by incubation 1 h at 37 degrees. GolgiPlug and GolgiStop (BD Biosciences) were then added according to manufacturer's instructions. After an additional 5 h of co-incubation at 37 °C, 5% C0₂, cells were transferred to V-bottom plates for FACS staining.

**Confocal microscopy**. FACS-sorted (same gating as above with addition of CD94/NKG2A positivity) or MACS-purified NK cells were used (purity verified >95% by same FACS panel as here above). NK cell negative selection kit was used for MACS-purification (Miltenyi). 10–50,000 NK cells were stained in V-bottom 96 well plates. DAPI was used for DNA stain of nucleus. The following primary antibodies were used perforin (dg9, BioLegend), Granzyme A (CB9, eBioscience), Granzyme B (GB11, BioLegend), Graunulysin (9/15 kDa, DH2, BioLegend, Pericentrin (Rabbit polyclonal, ab4448, Abcam), LAMP-1 (H4A3, BioLegend). Primary antibodies were detected using goat-anti-rabbit (for pericentrin) or goat anti-mouse isotype-specific antibodies conjugated to Alexa Fluor 488, 647, or 568.

Antibodies were incubated for 1 h at room temperature. Fixation was done with 2% PFA in PBS buffer. Permeabilisation was done with a 1% BSA, 0.4% saponin PBS buffer. Following staining and washing in V bottom plate, cells were transferred to 15-well µ-Slide Angiogenesis (ibidi) pre-coated with Cell-Tak (Corning). Slides were centrifuged for 2 min 300 rpm to let cells settle. Cells were then covered with ProLong Gold Antifade (ThermoFisher). Samples were imaged at room temperature with an Andor Revolution spinning disk system with CSU-X1 spinning disk (Yokogawa), 512 × 512, 16 µm² pixel camera (iXon, Andor) IX81 microscope (Olympus) using the 60× or 100× objective (numerical aperture 1.45) and 2.5× camera adaptor and with lasers exciting at 405, 488, 561, and 640 nm. Images were acquired using IQ2 (Andor) and processed using IMARIS software (Bitplane). The spots and surface functions were used to define structures between which distances were then measured using Imaris.

**Electron microscopy**. FACS-sorted NK cells (same gating as above with addition of CD94 positivity) were immediately fixed in 0.5% glutaraldehyde in 0.2M sodium cacodylate buffer (pH 7.2) for 30 min. They were then washed in sodium caco-dylate buffer, treated with reduced osmium tetroxide 1% OsO4, 1.5% potassium ferricyanide at room temperature for 60 min, washed in water, treated with 0.5% magnesium uranyl acetate at 4 °C for 16 h, dehydrated with ethanol rinsed in propylene oxide and embedded in Epon resin. Ultrathin sections were examined in a FEI Tecnai G2 TEM at 80 kV. Images were acquired with a MegaView III CCD and Soft Imaging Systems programme.

**KIR and HLA genotyping**. Genomic DNA was isolated from whole blood samples using the QIAamp DNA Mini Blood Kit (Qiagen). For decidua samples, the tissue was first digested using proteinase K and RNase A (Roche) in combination with tissue lysis and protein precipitation buffers (Qiagen). DNA was subsequently precipitated using isopropanol. Genotyping for KIR gene presence and HLA-C1/C2 status was performed by PCR-SSP using validated methods[22,23].

**Statistical analysis**. FCS files were analysed with FlowJo v10.5.3 (Tree Star Inc.). tSNE, DensVM clustering and nearest neighbour-based tissue probability analyses were performed using the R packages cytofkit and DepecheR from Bioconductor[69,70]. Statistical analyses were largely performed using PRISM (GraphPad Software Inc.) and the open source statistical package R (www.r-project.org). Datasets were tested for normal distribution and the appropriate statistical test was then used to compare subsets. Methods used are specified in figure legends.

**Reporting summary**. Further information on research design is available in the Nature Research Reporting Summary linked to this article.

## Data availability

The authors declare that the data supporting the findings of this study are available within the paper and its supplementary information file. The source data underlying Figs. 2C, 3A–D, 4A–D, 5A–I, 6A–D, and Supplementary Figs 1B, 2B, 2E–F, 5, 7, 8A–B, 9A–B, 10B are provided as Source Data file. Data not found in the source data are available upon request from the authors.

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

## Acknowledgements

We thank Mateusz Strzelecki and Richard Grenfell for assistance with CyTOF data acquisition at the flow cytometry core at Cancer Research UK Cambridge institute and Rachel Hipkin for her help in setting up the panel. We thank members of the Colucci and Moffett labs, past and present, for suggestions and discussions and Roser Vento-Tormo, Victoria Male and Laura Esposito for critical reading of the manuscript. We also thank all the staff and patients at Addenbrookes hospital, Cambridge without whom this study would not have been possible. This work was funded by the Wellcome Trust (Grant 200841/Z/16/Z to F.C. and A.M.), the Medical Research Council (Grant MR/P001092/1 to A.S.), the National Institutes of Health (Grants NIH U01 AI090905 and R01 AI17892 to P.P.), AstraZeneca-MedImmune, the Centre For Trophoblast Research (CTR) and the Cambridge NIHR BRC Cell Phenotyping Hub to F.C. O.H. was supported by a AstraZeneca-MedImmune-Cambridge PhD fellowship. M.A.I. was supported by The Wenner-Gren Foundation, Stockholm, Sweden.

## Author contributions

O.H., M.A.I., A.H., G.G., P.P., A.M., A.M.S., F.C. were involved in conceptualisation. O.H., M.A.I., L.G., M.H., A.M.S. performed the investigation. N.S. was involved in methodology. O.H., M.A.I., M.H., J.C.S., G.G., P.C., J.T., A.H., A.M., A.M.S., F.C. performed formal analysis. O.C. and L.E.F. performed KIR and HLA genotyping as part of investigation. H.G. provided resources. O.H., M.A.I., A.H., A.M., A.M.S., F.C. wrote the manuscript. A.M.S. and F.C. jointly supervised this work.

## Competing interests

H.G. (AstraZeneca) had a role in research funding and data interpretation. O.C. had a role in data interpretation when working for the University of Cambridge and moved to AstraZeneca during the preparation of the paper. Neither company representatives had a role in data selection for publication. All other authors declare no competing interests.

## Additional information

**Peer review information** *Nature Communications* thanks Brice Gaudilliere, Ofer Mandelboim, and other anonymous reviewers. Peer review reports are available.

