## [Peer Review File · Nature Communications]

Reviewers' comments:

Reviewer #1 (Remarks to the Author):

Using mass cytometry, Huhn and colleagues analyze human decidual innate lymphoid cells (dILCs). They provide evidence for multiple subsets of apparently mature decidual NK (dNK) cells as well as ILC3 and proliferating NK cell progenitors. These dILCs can be distinguished phenotypically from peripheral blood NK cells and further studies demonstrate functional differences in dNK subsets including cytokine and chemokine secretion patterns as well as expression of cytolytic machinery dependent on the type of stimulation (pharmacological activation versus missing self/K562). As dNK1 are the most abundant subset, the authors study the impact of KIR acquisition on the phenotype of these cells. As PB NK cells are known to acquire inhibitory KIR during 'education' that is associated with increased responsiveness, it was interesting to know how KIR expression impacted on dNK1 cells. The authors find that several markers on dNK cells increase with increasing KIR expression, including LILRB1, Ki67, NKp30 and granzyme B, while T-bet expression decreases. These changes are not observed in PB NK cells that are stratified by KIR expression, suggesting that PB and dNK cells may follow distinct 'education' pathways. Analysis of cytotoxic granules in dNK cells reveal that these structures are significantly larger and located further from the MTOC. In an effort to understand the biological relevance of differential KIR expression on dNK cells, the authors study cellular responses following KIR engagement. They find that dNK cell responsiveness is correlated with KIR acquisition with heightened responses to triggering by activating KIRs. In contrast, these same cells (essentially dNK1) show diminished responses to missing self again pointing to a differential regulation of NK cell responsiveness in PB and dNK cells.

The authors have made an extensive phenotypic analysis of human decidual ILCs and in particular, of human dNK cell subsets. They confirm several observations reported in a recent scRNA sequencing analysis (Vento-Tormo et al, 2018) concerning three major dNK cell subsets and the preferential KIR expression in dNK1. Their mass cytometric approach provides additional information about the heterogeneity of dNK cell subsets and further describes functional outputs (chemokine and cytokine production, granule protein expression) at the protein level that substantially extend the previous scRNAseq analyses. The observations concerning granule morphology and KIR activation of dNK cells are interesting and provide evidence for distinct regulation of dNK cells compared to their peripheral blood counterparts.

1) The authors identify 4 dNK1 subsets, 2 dNK3 subsets, dNK2 cells and a proliferating (putative) dNK cell precursor. It would be interesting to know how these different subsets are related in terms of development/differentiation. The authors use pharmacological activation to read out effector functions in dNK cells; are there any phenotypic changes that are induced by stimulation that help understand the relationships within dNK cell subsets (ie. the 4 dNK1 or 2 dNK3 cells) or between different dNK cell subsets?

2) The authors study several chemokine and cytokines following stimulation and present their data as percentages of cells expressing a given effector molecule. What about polyfunctionality in dNK cells? Are there dedicated subsets of cells that express single chemokines or are these cells capable of expressing multiple effectors at the same time? How does this correlate with expression of granule proteins? Does KIR expression alter the potential polyfunctionality of dNK cells ?

3) The granule phenotype in dNK cells is intriguing. The authors provide evidence that CD107a expression can be differentially regulated in KIR expressing dNK cells (Figure 6) depending on the activation stimulus. Is granule morphology differentially regulated in dNK cells upon PMA versus KIR2DS4 cross linking?

Reviewer #2 (Remarks to the Author):

The decidua is a maternal tissue which is present all along pregnancy, from the very beginning. It is heavily populated with immune cells which lay an important role in fetal development. In the manuscript titled "high dimensional analysis reveals tissue- specific granule organization and functional compartmentalization in human decidual innate lymphoid cells", Huhn et al. describe big data analysis of decidual mononuclear cells to identify different subpopulations inhabiting this maternal tissue. The authors use mass cytometry in order to get multi-dimensional information which shows heterogeneous cell populations of Natural Killer (NK) and Innate Lymphoid Cells (ILCs). The authors move on to elucidate phenotypic and functional differences between the populations, including response to activating stimuli, granule organization and cytokine secretion.

The paper revolves around an important subject which came to light in recent years: the maternal immune system during pregnancy and its effect on fetal development and pregnancy outcomes. The paper shows the complexity of the immune cell populations in the decidua and how each subpopulation differs in many aspects from other populations. That being said, there are a few issues which must be addressed:

1. Like many other papers which employ high dimensional analysis at the starting point, the manuscript is of rather descriptive nature. This requires careful writing so that the reader is able to follow the general story rather than drown in the details.
2. In Figure 3, the authors choose to use a radar plot in order to show population response to different stimuli. The purpose of graphs in any paper is to help the reader understand the written data better. It seems that in this case, the plots add a dimension of complication, rather than showing in a clear way what's written in the result section. The author should consider going with an alternative plot for data representation.
3. Although adding information about a fascinating subject, it is hard to see what the physiological relevance of the data in the paper is. The authors claim they have an organelle system in place, it seems that the different populations and their functions which were identified should be put to the test to show that the findings are relevant to the role these immune cells play during pregnancy. In conclusion, the manuscript revolves around a fascinating issue, yet the results seem a bit preliminary and require further work and proof of physiological relevance.

Reviewer #3 (Remarks to the Author):

Huhn et al. provide a comprehensive characterization of the cell phenotype, functional capacity, and morphological features of first trimester human decidual innate lymphoid cells (dILC) using a combination of mass cytometry, fluorescence flow cytometry and electron and confocal microscopy. From a technical standpoint, this study makes optimal use of mass cytometry, maximizing its read-out capacity by including phenotypic and functional markers necessary to capture NK cell subtype diversity. The number of samples (n=12) is acceptable for this type of exploratory, descriptive study. Taken together, this study provides a thorough insight into first trimester dILC heterogeneity, providing higher resolution than previous studies with similar objectives. It can serve as the basis and reference point to further explore dILC function for placentation and remodeling in future studies.

Notable limitations of the study include 1) the lack of clinical information for these samples, such as maternal age, parity, or co-morbidities and 2) the lack of evidence supporting the biological significance of the different dILC subtypes (ie corresponding functionality on placental cells such as EVT or dendritic cells).

Major comments

1- Provide gating strategy used to gate on major dILC clusters (Figure 3): Lin- cluster annotations are reported in Table 1. The description of the distinguishing features is vague, especially for the four newly described dNK1 subsets. This is crucial for the basic definition of dILC subtypes.

2- Responsiveness of cluster 7 (dNKp) to stimulation is – with the exception of KI67 - very similar to the excluded cluster 4 (unhealthy cells, Fig. S2E). Also, mean expression patterns seem similar in excluded c4 and included c7, except for Ki-67 (Fig. 2B). Could the authors please comment on the putative function of these non-responding, proliferating dNKp?

3- Cryopreserved vs. frozen samples

Please clarify when stains/analyses of cryopreserved vs. fresh cells are shown. What is shown in Fig. 2B: cryopreserved cells? 2C shows fresh stained cells. Thus, S2F is the 'true' donut plot for 2B?

4- General

Across the manuscript, it would be helpful to include information on ILC-specific markers for readers who are not ILC experts, eg. when introducing the ILC subsets for Fig. 2.

Minor comments

Figure 2:

- Thirteen dILC clusters are presented and condensed to the finally relevant populations in A-D. Figure 2 and the accompanying text could be reorganized, such that the order of the presented subtypes is as consistent as possible.

Figure 3:

- Please add short descriptions on attributes of cytokine/markers, when reporting results in text, eg. MIP-1b, preferentially produced by dNK2 and dNK3, does what?

Figure 4-6:

- Please indicate clearly in text and legend for each figure which dNK subtype (if any?) is used. Assuming from the possibility of having KIR=0, all dNK (Lin- CD56+) were used?

Figure 4:

- Please correct: PB n=8-9 in figure, vs. n=12 in legend text.

- Matched PB/decidua vs. unmatched donor PB: Have the authors checked if there is a difference in marker expression upon KIR acquisition in pregnant PB vs. unmatched donor PB? Some of the markers show variance that could be donor-related. Please provide n of pregnant vs (non-pregnant) unmatched donor in legend.

Point-by-point response to Reviewers

Dear Dr. Bondar, thank you for the opportunity to revise our manuscript in response to the reviewers' helpful comments. Please find below our point-by-point response. The changed text in the manuscript is highlighted in red. In particular we have sought to ensure the different ILC subsets analysed are more clearly defined for the reader throughout to make the manuscript more readable. We have slightly changed the title of the manuscript to "Distinctive Phenotypes and Functions of Innate Lymphoid cells in Human Decidua during early pregnancy" in response to one of the reviewers' comments concerned with clarity.

Reviewer #1

1. The authors use pharmacological activation to read out effector functions in dNK cells; are there any phenotypic changes that are induced by stimulation that help understand the relationships within dNK cell subsets (ie. the 4 dNK1 or 2 dNK3 cells) or between different dNK cell subsets?

We have re-analysed our data and observed a small decrease in CD96 expression upon PMA stimulation but not with K562. Because PMA is known to affect expression of other cell surface markers (Harrison 1991, Romee 2013), the significance of this change in CD96 is unclear for understanding relationships between dNK subsets. Based on the transcription factor staining, we have speculated in the Discussion (lines 355-358):

'we find that dNK1 are Eomes^{high}Tbet^{low}, in contrast to dNK3 which are Eomes^{int}Tbet^{high}. dNK2 express intermediate levels of both Eomes and Tbet suggesting that they might be positioned along a differentiation trajectory between dNK1 and dNK3'.

In response to reviewer 3, major comment 2, we have also added a new comment on the origin of dNKp (lines 168-171):

' dNK are known to proliferate in vivo and based on their expression of moderate levels of KIR and NKG2A and low expression of CD117 and CD127, we believe the Ki-67+ dNKp population represent a mixture of dNK cells, including dNK1, that are dividing within decidual tissue (King 1991)'.

2. What about polyfunctionality in dNK cells? Are their dedicated subsets of cells that express single chemokines or are these cells capable of expressing multiple effectors at the same time?

We have re-analysed the data in revised Fig 3 to measure polyfunctionality. Fig 3B legend and the text in the results section have been amended accordingly on lines 233-241:

'The ability of each dNK subset to mount a polyfunctional response to stimulation was assessed by examining three functional readouts: CD107a staining or IFN γ or at least one of the cytokines MIP1 α , MIP1 β , GM-CSF, XCL1 (Fig.3B). dNK3 showed the strongest polyfunctional responses; more than 75% upregulating at least two readouts simultaneously, with 37% of cells expressing all three. In contrast, 60% of dNK1 expressed a single readout, most of which were CD107a- IFN γ -, indicating these cells are specialised for cytokine secretion, other than IFN γ . While dNK2 displayed intermediate polyfunctionality, all three dNK subsets displayed distinctive patterns of polyfunctional responses compared to pbNK-like cells.'

We have also amended the methods section accordingly on lines 593-600.

How does this correlate with expression of granule proteins? Does KIR expression alter the potential polyfunctionality of dNK cells?

See answer to question 3, below.

3) Is granule morphology differentially regulated in dNK cells upon PMA versus KIR2DS4 cross linking?

Measuring changes in granule morphology during the short duration of stimulation by PMA or 2DS4 cross-linking would be difficult, but we have discussed the possibility that differences in granule organisation between pbNK and dNK may account for the differing responses to stimulation with increasing KIR. We highlight this on lines 315-317. We have also sought to clarify this point by adding in the discussion, lines 418-419:

'The different functional responses of dNK and pbNK as they acquire more KIR, may be due to the differences observed in granule organisation between the two.'

When considering whether KIR expression alters potential polyfunctionality of dNK, we show in Figure 6, that as number of KIR increases, XCL1 staining and degranulation both decrease when stimulated in K562 assay, but do not change with increasing KIR after PMA stimulation. In contrast, with 2DS4 cross linking, both XCL1 and degranulation increases as number of KIR expressed increases. This means a higher proportion of dNK express both XCL1 and CD107 after 2DS4 cross linking in dNK with 3 KIR compared to dNK with 1 KIR- demonstrating increasing polyfunctionality with increasing KIR. However this is only true for 2DS4 X-linking and not for K562 stimulation. Thus, the effect of KIR on dNK polyfunctionality varies depending on the stimulus used. We hope that the discussion of this in lines 338-339 now makes this clear:

'A higher proportion of dNK express both readouts as the number of KIR expressed goes up, so polyfunctionality of dNK correlates with KIR co-expression (data not shown).'

We would be happy to provide this data as a Supplementary figure if the reviewer feels more than the above statement is required, in addition to the data already shown in figure 6C.

Reviewer #2

1. Like many other papers which employ high dimensional analysis at the starting point, the manuscript is of rather descriptive nature. This requires careful writing so that the reader is able to follow the general story rather than drown in the details.

We have sought to improve clarity and readability throughout for the general reader. This includes more clearly defining the phenotype of individual ILC subsets. We have extensively revised the paragraph "Characterisation of decidual ILC subsets" (lines 156-184) and Table 1, in which we now carefully describe the phenotype used to define the different clusters identified. We have also revised Figure 3 (see below), which now represents data in a simplified way. We have also opted for the simplified title *'Distinctive Phenotypes and Functions of Innate Lymphoid cells in Human Decidua during early pregnancy'*.

2. In Figure 3, the authors choose to use a radar plot in order to show population response to different stimuli. The purpose of graphs in any paper is to help the reader understand the written data better. It seems that in this case, the plots add a dimension of complication, rather than showing in a clear way what's written in the result section. The author should consider going with an alternative plot for data representation.

The data originally shown in Figure 3B as Radar plots of responses to PMA/ionomycin is now presented as scatter plots (see new Fig S7). The corresponding radar plots of responses of the ILC subsets to K562 stimulation (original Fig S7), have now been amended in the same way (now Fig S8). The text in the manuscript has been altered to reflect these

changes. Fig3B has now been changed to include an analysis of polyfunctional responses by dNK subsets as requested also by Reviewer #1 – point 2. The results of the new polyfunctionality analysis in Fig 3B, are described in lines 233-241. For methods used of this new analysis, see lines 593-600.

3. In conclusion, the manuscript revolves around a fascinating issue, yet the results seem a bit preliminary and require further work and proof of physiological relevance.

To clarify the rationale for the careful phenotypic and functional analysis of decidual ILCs we have carried out, we have added at lines 103-10:

'ILCs exhibit considerable heterogeneity and flexibility to differentiate from one type to another, necessitating detailed characterization within each tissue type (Vivier et al., 2018).'

We agree that there is little evidence directly supporting the biological significance of dILC subtypes, however in this manuscript we have properly characterised their heterogeneity and shown unique and unusual properties in relation to education, functional capacity and organisation of granules. We believe this provides an essential basis for future studies on their function in both pregnant decidua and in endometrium. To emphasise the future physiological relevance, we have added the following to the discussion (lines 475-477):

'This characterization of the unique features of decidual ILCs, will now permit future studies of their functional interactions with other decidual cells as well as fetal trophoblast.'

We speculate on the possible roles of dILCs in the final part of the Discussion (lines 461-479), where we suggest that further research involving 3D organoid culture systems derived from both maternal endometria and placentas will be instrumental in addressing the roles of decidual ILC subsets.

Reviewer #3

Notable limitations of the study include: 1) the lack of clinical information for these samples, such as maternal age, parity, or co-morbidities; 2) the lack of evidence supporting the biological significance of the different dILC subtypes (ie corresponding functionality on placental cells such as EVT or dendritic cells).

1. 1) Unfortunately, the ethical permission for this study does not allow collection of data on maternal age or parity. We now clarify in Methods (lines 508-510) that only healthy women with normal pregnancies were recruited for these studies.
2. 2) We agree that there is little evidence supporting the biological significance of dILC subtypes, however we have properly characterised their heterogeneity and shown unique and unusual properties in relation to education, functional capacity and organisation of granules. (See also response above to reviewer 2, comment 3). The possible roles of dILCs is speculated in the final part of the Discussion, where we suggest that further research involving 3D organoid culture systems derived from both maternal endometria and placentas will be instrumental in addressing the roles of decidual ILC subsets. We believe the study reported here provides an essential basis for future studies on their function in both pregnant decidua and in endometrium.

Major comments

1- Provide gating strategy used to gate on major dILC clusters (Figure 3): $Lin^{cluster}$ annotations are reported in Table 1. The description of the distinguishing features is vague,

especially for the four newly described dNK1 subsets. This is crucial for the basic definition of dILC subtypes.

We now provide in Table 1 the detailed phenotypes used to identify dILC clusters and clarify the gating strategy used to identify the major ILC clusters analysed in Figure 3. See lines 222-223:

'We generated a new tSNE map after dNK stimulation and used this to gate manually on the Lin-CD56+ subsets defined in Fig. 2 and Table 1'.

2- Responsiveness of cluster 7 (dNKp) to stimulation is – with the exception of Ki67 - very similar to the excluded cluster 4 (unhealthy cells, Fig. S2E). Also, mean expression patterns seem similar in excluded c4 and included c7, except for Ki-67 (Fig. 2B). Could the authors please comment on the putative function of these non-responding, proliferating dNKp?

We suggest that the Ki67+ 'dNKp' population represent a mixture of dNK cells, including dNK1, that are proliferating within decidual tissue. To clarify this, we have added the following on lines 168-174:

'dNK are known to proliferate in vivo and based on their expression of moderate levels of KIR and NKG2A and low expression of CD117 and CD127, we believe the Ki-67+ dNKp population represent a mixture of dNK cells, including dNK1, that are dividing within decidual tissue (King 1991). dNK1 (c10-13) are the most abundant subset identified in cryopreserved dILCs (30%), followed by dNK2 (c9, ~15%) and dNK3 (c5,c8, ~15%) (Fig. 2C). The proportion of dNK1 in freshly isolated samples was even greater than for cryopreserved samples (fig. S2F).'

3- Cryopreserved vs. frozen samples Please clarify when stains/analyses of cryopreserved vs. fresh cells are shown. What is shown in Fig. 2B: cryopreserved cells? 2C shows fresh stained cells. Thus, S2F is the 'true' donut plot for 2B?

We agree that this was confusing. In order to show that cluster 4 cells are a feature of cryopreservation we have compared analysis of freshly isolated and frozen cells. Fig 2A and 2B show results from cryopreserved cells and original Fig 2C was from freshly stained cells. Fig 2C has now been changed so that it also shows data from cryopreserved cells. Instead Fig S2F, now shows the proportions of each cluster in freshly isolated cells for comparison to the results in Fig 2C. The paper has been amended on lines 171-174, and in legends to Fig 2C and Fig S2F to explain this:

'dNK1 (c10-13) are the most abundant subset identified in cryopreserved dILCs (30%), followed by dNK2 (c9, ~15%) and dNK3 (c5,c8, ~15%) (Fig. 2C). The proportion of dNK1 in freshly isolated samples was even greater than for cryopreserved samples (fig. S2F).'

4- General

Across the manuscript, it would be helpful to include information on ILC-specific markers for readers who are not ILC experts, eg. when introducing the ILC subsets for Fig. 2.

We have added additional information to clarify how the various ILC and NK subsets were identified in Fig 2, and to clarify to the reader the phenotypic characteristics of the ILC subsets. See Table 1 and in lines 156-184. See also response to reviewer 2, point 1 above.

Reviewer 3, Minor comments

1. *Figure 2: - Thirteen dILC clusters are presented and condensed to the finally relevant populations in A-D. Figure 2 and the accompanying text could be reorganized, such that the order of the presented subtypes is as consistent as possible.*

As the reviewer suggests, we have now amended Figs 2C and D, to ensure the order of subsets is presented in a more consistent way (reading dNK1,2,3 pbNK, ILC3 etc.). The order in Fig 2B is determined by the dendrogram so cannot be altered. We have revised the accompanying text to clarify the phenotypes of the presented subtypes (lines 156-184 and Table 1).

2. *Figure 3:*

- Please add short descriptions on attributes of cytokine/markers, when reporting results in text, eg. MIP-1b, preferentially produced by dNK2 and dNK3, does what?

We have added the following on lines 213-221:

'GM-CSF (Granulocyte-macrophage colony-stimulating factor) stimulates haematopoietic stem cells to produce granulocytes and monocytes, but more relevant to the decidua is its ability to attract EVT (Xiong 2013). The role of IFN γ in human decidua is unclear, whereas it is an essential cytokine for vascular remodelling in murine decidua (Ashkar 2000). XCL1 (lymphotactin), is believed to act on specific antigen-presenting DC subsets in tissues and the receptor is also expressed on EVT (Kennedy 2016). MIP1 α (Macrophage inflammatory protein-1 α) binds CCR1, CCR4 and CCR5 and MIP1 β binds CCR5 and CCR8; in other tissues they have been shown to recruit various leukocytes, including NK cells, neutrophils and monocytes.'

3. *Figure 4-6:*

- Please indicate clearly in text and legend for each figure which dNK subtype (if any?) is used. Assuming from the possibility of having KIR=0, all dNK (Lin- CD56+) were used?

As the reviewer has suggested we have added lines (294 and 325) to clarify that these figures were generated using Lin- CD56+ dMCs. These analyses explicitly address the effect of KIR acquisition and are not directly stratified by dNK subset. However, KIR+ cells will be largely comprised of dNK1 whilst KIR- cells will be other dILC. For this reason, our statistical analysis only examined changes within the KIR+ dNK in Fig 4 (i.e. KIR1,2,3+). To be accurate, we have changed the abstract from dNK1 to 'dNK have distinctive organisation and content...'

4. *Figure 4:*

- Please correct: PB n=8-9 in figure, vs. n=12 in legend text.

Corrected – see legend to Fig 4D which now says n=8-9, (line 750).

- Matched PB/decidua vs. unmatched donor PB: Have the authors checked if there is a difference in marker expression upon KIR acquisition in pregnant PB vs. unmatched donor PB? Some of the markers show variance that could be donor related. Please provide n of pregnant vs (non-pregnant) unmatched donor in legend.

Among the 8-9 unmatched blood donors in Fig 4, Five are pregnant and 4 non-pregnant (of the latter, 3 donors are used in some experiments and 4 in others). Although there might be donor-related variance, the sample size is too small to draw any conclusions.

REVIEWERS' COMMENTS:

Reviewer #1 (Remarks to the Author):

The authors have adequately addressed my comments.

Reviewer #3 (Remarks to the Author):

Thank you for the opportunity to review a revised version of the manuscript by Huhn et al. describing a comprehensive and functional characterization of decidual ILC subsets during a human pregnancy. The authors have addressed all my comments and the revised manuscript is much improved. I have no further comments.